# Perceived Kinesiophobia and Its Association with Return to Sports Activity Following Anterior Cruciate Ligament Reconstruction Surgery: A Cross-Sectional Study

**DOI:** 10.3390/ijerph191710776

**Published:** 2022-08-30

**Authors:** Abdullah Raizah, Ali Alhefzi, Ahmad Ayed M Alshubruqi, Majed Abdullah M. Al Hoban, Irshad Ahmad, Fuzail Ahmad

**Affiliations:** 1Department of Orthopedics, College of Medicine, King Khalid University, Abha 62529, Saudi Arabia; 2Department of Emergency Medicine, Asser Central Hospital, Abha 62523, Saudi Arabia; 3Department of Medical Rehabilitation Sciences, College of Applied Medical Sciences, King Khalid University, Abha 62529, Saudi Arabia; 4College of Applied Sciences, AlMareefa University, Riyadh 13713, Saudi Arabia

**Keywords:** ACL, fear, kinesiophobia, reconstruction, physical activity, knee, instability

## Abstract

**Background:** The knee joint is one of the most important joints in terms of its functions of providing great stability, movement and weight bearing. Among the contributors to knee joint stability, there is the anterior cruciate ligament (ACL). Kinesiophobia is said to be the fear of movement or the fear of re-injury. Kinesiophobia is the most extreme form of fear of movement, and it is defined as an excessive, irrational, and debilitating fear of physical movement and activity resulting from a feeling of vulnerability to painful injury or re-injury. **Aim:** To estimate the prevalence and effect of kinesiophobia among patients with ACL reconstruction in the Aseer region, in southern Saudi Arabia. **Methodology:** A descriptive cross-sectional approach was used involving those patients who underwent ACL reconstruction surgery in Aseer Central Hospital during the period of October 2017 to October 2019. The Tampa Scale for Kinesiophobia (TSK) and ACL—Return to Sport after Injury (ACL-RSI) scale were used to determine kinesiophobia and the readiness to return to sport after ACL injury or reconstructive surgery. **Result:** The research included 130 ACL reconstruction patients with ages ranging from 18 to 45 years with a mean age of 27.2 + 7.5 years. More than 97% of the participants were males. In 67.7% of the cases, the right leg was affected. A total of 10.8% of the patients recorded a low level of kinesiophobia, while only 6.9% recorded a high level. **Conclusions:** In conclusion, the study revealed that among patients who underwent ACL reconstruction, kinesiophobia was at a moderate level. Kinesiophobia was recorded more among middle-aged patients who waited a long time from the onset of injury until the ACL reconstruction surgery time.

## 1. Introduction

After a knee injury, the anterior cruciate ligament (ACL) must be surgically replaced using tissue grafts to restore its function [1]. If the torn ACL is surgically removed from the knee joint during arthroscopic ACL restoration, the graft may be passed through the remaining ACL tissue [2]. In 1994, there were roughly 33 ACL injuries for every 100,000 people; in 2014, there were between 40 and 60 incidences for every 100,000 people. Individuals in sports-related occupations are more vulnerable to the effects of ACL injuries [3,4].

Injuries to the ACL may be divided into two classes, contact and non-contact, according to the way the damage has occurred. Whenever a person or object forcefully impacts the knee, this may cause the ligaments in the knee to rip, causing contact injury [5]. Decelerating, cutting, or landing from a leap are all common actions that may cause a non-contact tear [5]. The incidence of torn ACLs in female athletes is three times greater than in male athletes [6]. Some of the factors that contribute to the variation in ACL rupture rates between men and women include a higher Q angle and hormonal variances [7,8].

Anxiety produced by the fear of experiencing pain as a result of movement is known as kinesiophobia [9]. In addition to slowing the recovery process, kinesiophobia hinders rehabilitation and prolongs disability and pain [10,11,12]. Fear and other negative or maladaptive psychological elements are among the most significant biopsychosocial factors that influence pain [13,14].

In the field of sports research, one area that is now undergoing development is the creation of profile models, which will assist in determining the outcome of therapy for ACL injuries. The capacity to identify patients who are at a greater risk of acquiring long-term chronicity is a vital component to effectively stratifying healthcare resources and maximizing treatment results. The goal of this research was to establish the amount of kinesiophobia among athletes who had had ACL reconstruction, as well as its connection with returning to physical activity, in order to determine the effectiveness of therapy.

## 2. Materials and Methods

### 2.1. Study Design

A descriptive cross-sectional approach was used.

### 2.2. Ethical Consideration

The study was conducted in accordance with the Declaration of Helsinki and is approved by the ACH Ethics and IRB (Internal Review Board) Committee of King Khalid University with REC #2017-02-42 (16 March 2O17), in accordance with departmental and hospital policy and with the fact that the facilities and resources specified in the protocol are adequate and available for studies involving humans.

### 2.3. Study Participants and Setting

The eligibility criteria for inclusion in the research were patients who had single ACL repair surgery between six months and up to two years prior to the commencement of the study. Patients with a history of multiple knee injuries or any other condition that would preclude them from engaging in sports were not included in the study.

All the patients who underwent ACL reconstruction surgery in Aseer Central Hospital during the period of October 2017 to October 2019 and fulfilled the inclusion criteria were recruited for this study.

### 2.4. Procedures

Informed consent was obtained from all the eligible participants. Data were collected using a pre-structured, self-administered questionnaire that covered the patient’s socio-demographic, ACL injury, and surgery data.

The ACL—Return to Sport after Injury (ACL-RSI) scale was used to determine the psychological readiness to return to sport after ACL injury or reconstructive surgery. This 12-item unidimensional measure examined emotions, performance confidence, and risk assessment after returning to sport following an athletic injury [15]. Each of the items of the scale was assessed on a visual analogue scale ranging from 0 (completely negative psychological reactions) to 100 (no negative psychological responses), with a higher score indicating a greater psychological preparedness to return to sports [16].

Kinesiophobia was assessed using the Tampa Scale for Kinesiophobia (TSK) [17], a questionnaire consisting of 17 questions that elicited respondents’ subjective assessments of various fear-related ideas. The Likert scale that was used for scoring ranged from “strongly agree” to “strongly disagree”, with a total score ranging from 17 to 68 and higher scores indicating greater degrees of fear avoidance behavior. TSK has shown a strong inter-tester reliability and concept validity, and it is currently frequently used in many clinical settings to identify possibly kinesiophobic behavior.

### 2.5. Statistical Methods

The data were extracted, revised, coded, and fed to statistical software IBM SPSS Statistics for Windows, Version 27.0, IBM Corp., Armonk, NY, USA. A descriptive analysis based on the frequency and percent distribution was conducted for all patients’ demographic and injury-related data. As for the kinesiophobia scale, all discrete items were summed after reversing the scores of items 4, 8, 12, and 16. The total score was categorized into three levels, a low level for those who had a score of 17–34 points, a moderate level for those with a score of 35–52 points, and a high level for those with a score of 53–68 points. The bivariate relation between patients and injury data was tested using the Mont Carlo Estimate and Exact probability test. A correlation analysis was conducted between the kinesiophobia level and ACL-RSI scale after reversing the degrees, in order to detect the nature of the relation.

## 3. Results

The research included 130 ACL reconstruction patients between the ages of 18 to 45 years, with a mean age of 27.2 + 7.5 years. More than 97% of the participants were males, and 53.3% were either overweight or obese (Table 1).

With regard to the ACL injury data (Table 2), the right leg was the affected side in 67.7% of the cases. About 9% of the cases had reconstruction surgery directly after injury, and 46.2% had the reconstruction within six months to one year, while 19.2% had the surgery after one year. A total of 32.3% of patients had surgery within the previous year, whereas 30.8% of cases had undergone ACL reconstruction surgery more than 18 months ago.

Table 3 shows the description of patients’ answers regarding kinesiophobia items. 67.7% of patients agreed that no one should have to exercise when he/she is in pain. In addition, 57.7% of patients felt that there is no risk in exercising even when there is pain. Patients agreed on numerous points, including the view that being physically active in the presence of pain would be beneficial, their concern for self-injury while exercising, and the fact that it is not safe for individuals who have undergone knee reconstruction to engage in exercises. About 32% of patients think that if they strive to overcome their fear, they will feel more pain. A low level of kinesiophobia was found in 10.8% of patients, whereas a severe level was seen in 6.9%.

There was a statistically significant difference (*p* = 0.001) between participants of different age groups, with 27.3% of participants in the age group of 35–45 years reporting a higher level of kinesiophobia.

As for the duration till surgery (Table 4), 10% of participants who had surgery after 6–12 months showed a higher level of kinesiophobia when compared to those who underwent the ACL reconstruction immediately or after one year (*p* = 0.001).

Compared to the patients who underwent ACL reconstruction surgery less than a year or two years post-surgery, 18.8% participants showed a higher degree of kinesiophobia between 1–2 years following ACL repair. This difference was found to be statistically significant (*p* = 0.001). All other factors including gender, obesity, and injury site were not related with kinesiophobia.

To find out the association between kinesiophobia and a return to sports activity (Table 5), the study revealed that there was a significant positive intermediate correlation between all five items associated with emotions. The items related to being nervous about playing one’s sports (r = 0.22), being frustrated about having to consider one’s knee with respect to one’s sport (r = 0.28), fear of re-injury by playing one’s sport (r = 0.17) and fear of accidentally injuring one’s knee by playing sports (r = 0.31) showed a positive correlation (Table 5). All other items showed no correlation with kinesiophobia.

## 4. Discussion

Musculoskeletal diseases, such as ACL reconstruction surgery, require several considerations when designing a rehabilitation program for individuals suffering from them. Though it is apparent that physical profiles are a key part of the rehabilitation protocols, the patient’s emotional and cognitive reactions to the damage should also be considered [18]. While its significance may be evident, kinesiophobia or fear of movement, which is a psychological reaction to injury, is difficult to evaluate. Early theories of fear-avoidance argued that the avoidance was not based on physical pain, but rather on more contextual variables linking pain to fear of injury [19].

The purpose of our research was to find out to what extent post-ACL reconstruction kinesiophobia affects patients’ desire to return to their previous level of physical activity. The study revealed that the majority of patients had a moderate level of kinesiophobia and that those with a high level did not exceed 6% of cases. Participants gave high scores to criteria such as fear of re-injury, inadvertent harm, pain aggravation, and being watchful of inappropriate movements. A possible explanation is that the patients included in the study were all under 30 years old, indicating that they need regular physical activity and are not content to remain bedridden or otherwise immobile for long periods of time. A previous study examined the relationship between kinesiophobia and the activity level and discovered that individuals with a lower level of exercise may have higher ratings six to twelve months following surgery [20].

This is more apparent from the fact that the highest level of kinesiophobia was recorded among those aged above 35 years. The study also revealed that kinesiophobia was higher among those who waited long until surgery, and this can be explained by their fear of repeating the situation of waiting for a long time in case they are re-injured. Patients who underwent ACL reconstruction a long time ago had a low level of kinesiophobia, which gave them a sense of safety due to almost completely healing.

With regard to the effect of kinesiophobia on their preparedness to return back to physical activity, the study proved the direct effect of kinesiophobia on their psychological mode as they were more nervous and afraid of re-injury. These findings were concordant with other studies that concluded that kinesiophobia was an important factor that affected people not returning to their pre-injury level or to full functional activity because of it [21].

The psychological aspects of recovery play a critical role in functional outcomes, and the relationship between these markers and kinesiophobia in relation to returning to sport activity after ACL reconstruction has only been studied in a small number of studies [22,23]. Furthermore, there are no findings in the literature on the functional differences between individuals with different levels of kinesiophobia and the return to sport after ACL repair. In this study, we discovered a link between higher levels of kinesiophobia and the return to sport based on the biopsychosocial factors that influence pain.

In order to effectively care for patients following anterior cruciate ligament restoration, especially those who want to return to sport, a deeper knowledge of the psychological components of rehabilitation is required. There should be more research into how kinesiophobia and confidence relate to objective measurements of strength and function. In addition, further research is required to examine the unknown cause-and-effect link between these factors and to see whether boosting one’s strength and performance in advance of returning to sport will help alleviate kinesiophobia.

Prior to returning to sports activities, it may be worthwhile to conduct an assessment for kinesiophobia, as it may be linked to future activity and functional performance as well as to secondary injury rates. Future studies should build on these preliminary findings to better understand the role of kinesiophobia and other psychological factors in decision-making about returning to sports activities, as well as in the possibility for rehabilitation to have a positive effect.

### Limitations of the Study

Participants in the study were mostly young and active. Although this sample represents a high-risk group, it may not reflect all patients who receive ACLR. Therefore, the generalizability of this cohort to other demographics, such as older patients or those having a sedentary lifestyle, may be limited.

## 5. Conclusions

In conclusion, the study revealed that kinesiophobia among patients who had ACL reconstruction was at a moderate level, affecting their preparedness to return to a pre-surgery physical activity level. Kinesiophobia was recorded more among middle-aged patients who waited a long time between the onset of injury and the surgery time. Researchers recommend that patients with ACL reconstruction receive a post-surgical rehabilitation program including psychosocial support in addition to the physical components.

## Figures and Tables

**Table 1 ijerph-19-10776-t001:** Personal data of patients undergoing ACL reconstruction in Aseer Central Hospital, Saudi Arabia.

Personal Data	No	%
*Age in years*		
15–24	54	41.5%
25–34	43	33.1%
35–45	33	25.4%
*Gender*		
Male	127	97.7%
Female	3	2.3%
*Body mass index*		
Normal	60	46.2%
Overweight	60	46.2%
Obese	10	7.7%
*Dominant leg*		
Right	85	65.4%
Left	45	34.6%

**Table 2 ijerph-19-10776-t002:** Injury data of patients undergoing ACL reconstruction in Aseer Central Hospital, Saudi Arabia.

ACL Injury Data	No	%
*Injury site*		
Right	88	67.7%
Left	42	32.3%
*Injury at dominant leg*		
Yes	91	70.0%
No	39	30.0%
*Duration between injury and reconstruction*		
At the same time	12	9.2%
1–5 months	33	25.4%
6–12 months	60	46.2%
>12 months	25	19.2%
*Duration after reconstruction*		
<1 year	42	32.3%
1–1.5 years	48	36.9%
1.5–2 years	40	30.8%

**Table 3 ijerph-19-10776-t003:** Agreement rate for different items of kinesiophobia among patients undergoing ACL reconstruction in Aseer Central Hospital, Saudi Arabia.

Kinesiophobia Item	Agreement Rate [%]
I’m afraid that I might injure myself if I exercise	63 [48.5%]
If I were to try to overcome it, my pain would increase	42 [32.3%]
My body is telling me I have something that is dangerously wrong	51 [39.2%]
My pain would probably be relieved if I were to exercise	64 [49.2%]
People aren’t taking my medical condition seriously enough	64 [49.2%]
My accident has put my body at risk for the rest of my life	60 [46.2%]
Pain always means that I have injured my body	57 [43.8%]
Just because something aggravates my pain does not mean that it is dangerous	57 [43.8%]
I am afraid that I might injure myself accidentally	66 [50.8%]
Simply being careful not to make any unnecessary movements is the safest thing I can do to prevent my pain from worsening	66 [50.8%]
I wouldn’t have this much pain if there weren’t something potentially dangerous going on in my body	70 [53.8%]
Although my condition is painful, I would be better off if I were physically active	66 [50.8%]
Pain lets me know when to stop exercising so that I don’t injure myself	52 [40.0%]
It’s really not safe for a person with a condition like mine to be physically active	66 [50.8%]
I can’t do all the things normal people do because it’s too easy for me to get injured	60 [46.2%]
Even though something is causing me a lot of pain, I don’t think it’s actually dangerous	75 [57.7%]
No one should have to exercise when he/she is in pain	88 [67.7%]

**Table 4 ijerph-19-10776-t004:** Factors associated with kinesiophobia level among patients undergoing ACL reconstruction in Aseer Central Hospital, Saudi Arabia.

Factors		Kinesiophobia Severity						*p*-Value
		Low [*n* = 14]		Moderate [*n* = 107]		High [*n* = 9]		
		No	%	No	%	No	%	
*Age in years*	15–24	7	13.0%	47	87.0%	0	0.0%	0.001 *
	25–34	1	2.3%	42	97.7%	0	0.0%	
	35–45	6	18.2%	18	54.5%	9	27.3%	
*Gender*	Male	14	11.0%	104	81.9%	9	7.1%	0.719
	Female	0	0.0%	3	100.0%	0	0.0%	
*Body mass index*	Normal	7	11.7%	50	83.3%	3	5.0%	0.729
	Overweight	6	10.0%	48	80.0%	6	10.0%	
	Obese	1	10.0%	9	90.0%	0	0.0%	
*Injury at dominant leg*	Yes	11	12.1%	74	81.3%	6	6.6%	0.751
	No	3	7.7%	33	84.6%	3	7.7%	
*Duration between injury and reconstruction*	At the same time	3	25.0%	9	75.0%	0	0.0%	0.001 *
	1–5 months	1	3.0%	29	87.9%	3	9.1%	
	6–12 months	0	0.0%	54	90.0%	6	10.0%	
	>12 months	10	40.0%	15	60.0%	0	0.0%	
*Duration after reconstruction*	<1 year	6	14.3%	36	85.7%	0	0.0%	0.001 *
	1–2	1	2.1%	38	79.2%	9	18.8%	
	>2 years	7	17.5%	33	82.5%	0	0.0%	

*p*: Mont Carlo exact probability; * *p* < 0.05 [significant].

**Table 5 ijerph-19-10776-t005:** Correlation between kinesiophobia and ACL-RSI scores among patients undergoing ACL reconstruction in Aseer Central Hospital, Saudi Arabia.

Return to Sport Items	ACL-RSI Score	R
Are you nervous about playing your sport?	43%	0.22 *
Do you find it frustrating to have to consider your knee with respect to your sport?	44%	0.28 **
Do you feel relaxed about playing your sport?	40%	0.147
Are you fearful of re-injuring your knee by playing your sport?	39%	0.17 *
Are you afraid of accidentally injuring your knee by playing your sport?	45%	0.31 **
Are you confident that your knee will not give way by playing your sport?	33%	−0.041
Are you confident that you could play your sport without concern for your knee?	31%	0.137
Are you confident about your knee holding up under pressure?	47%	0.021
Are you confident that you can perform at your previous level of sport participation?	37%	−0.054
Are you confident about your ability to perform well at your sport?	41%	−0.005
Do you think you are likely to re-injure your knee by participating in your sport?	42%	0.157
Do thoughts of having to go through surgery and rehabilitation prevent you from playing your sport?	44%	0.063

R: Pearson correlation coefficient; *: *p* < 0.05; **: *p* < 0.001.

## Data Availability

The dataset for the result of this study will be available from the corresponding author upon reasonable request.

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
