# Peer review of "Perceived Kinesiophobia and Its Association with Return to Sports Activity Following Anterior Cruciate Ligament Reconstruction Surgery: A Cross-Sectional Study"

_ijerph, 2022, doi:10.3390/ijerph191710776_

Round 1
Reviewer 1 Report
This is a simple cross-sectional study that investigates kinesiophobia in a population who undergone ACL reconstruction surgery. Although the idea behind this study is nice, there are many points that must be improved in this manuscript and in the way data are presented. A list including some minor and major points follows.
- the word kinesiophobia is the most important one according to the Abstract, and it recurs in the whole manuscript, although it does not appear in the title. the title should be more accurate.
- the first paragraph (line 33-38) is not justified by the content of the rest of the article. in addition, it is not supported by any citation. therefore, I suggest to delete it completely.
- line 76-77 is not clear from a syntactical point of view. what is the meaning of “for 6 months up to 2 years”? from when?
- “RSI scale” (line 80): what does RSI stand for? it is better to explain acronyms in the text. It is also important to describe the characteristics of the scale and questionnaires that are being used, i.e., in this case, which are the “individual items” of the RSI scale? (line 82). regarding this, the paragraph about the TSK is correct.
- cit. number 15 and 16 have numbers 41 and 40 in it (delete them).
- in the Methodology paragraph in the abstract, specify the questionnaires and biometric tests used in the study.
- “study participant and setting” paragraph: are there any inclusion criteria based on age? the “setting” is not described (which hospital, etc.). on the other hand, questionnaires are not either “participant” or “setting”, so I suggest to put every test made on the study population in another paragraph. I also suggest to better explain the “study design”, i.e., for example: cross-sectional study assessing the level of kinesiophobia through questionnaires and the ability to move through biometrical assessments.
- in Table 1, the first age of participants is 15 years old, although in the text it is specified that the study sample’s age ranged from 18 to 45.
- line 135 is grammatically incorrect.
- line 153: “In comparison to the patients who had the ACL reconstruction surgery within the 153 year or after two years post-surgery “ - why post-surgery? please revise this phrase syntactically
- results: what about the RSI scale? what about the biometric tests specified in the “procedures” paragraph? if the results of these tests do not appear in the “results” paragraph, it is useless to describe them in the Methodology section.
- discussion: “In this study, we are the first to discover a link between higher levels of Kinesiophobia and return to sport based on their functional performance disparities” (line 201). that is incorrect. the present study did not investigate anything related to functional performance - at least it was not explained in the Results section.
- at the beginning of the manuscript you also cited the difference between a direct and an indirect injury to the ACL. I am curious whether you considered that piece of information in your study. does the amount of kinesiophobia increase according to the type of injury?
Author Response
Comments and Suggestions for Authors
This is a simple cross-sectional study that investigates kinesiophobia in a population who undergone ACL reconstruction surgery. Although the idea behind this study is nice, there are many points that must be improved in this manuscript and in the way data are presented. A list including some minor and major points follows.
- the word kinesiophobia is the most important one according to the Abstract, and it recurs in the whole manuscript, although it does not appear in the title. the title should be more accurate.
Author’s Response: As suggested by the reviewer the tittle has been revised to “Perceived Kinesiophobia and Its Association with Return to Sports Activity Following Anterior Cruciate Ligament Reconstruction Surgery: A Cross‐Sectional Study.”
-The first paragraph (line 33-38) is not justified by the content of the rest of the article. in addition, it is not supported by any citation. therefore, I suggest to delete it completely.
Author’s Response: Line 33-38 has been deleted
-Line 76-77 is not clear from a syntactical point of view. what is the meaning of “for 6 months up to 2 years”? from when?
Author’s Response: The duration was in context with the date of surgery. The statement has been clarified.
-“RSI scale” (line 80): what does RSI stand for? it is better to explain acronyms in the text. It is also important to describe the characteristics of the scale and questionnaires that are being used, i.e., in this case, which are the “individual items” of the RSI scale? (line 82). regarding this, the paragraph about the TSK is correct.
Author’s Response: The paragraph related to ACL-RSI was revised and additional details about the scale was provided.
-Cit. number 15 and 16 have numbers 41 and 40 in it (delete them).
Author’s Response: deleted
-In the Methodology paragraph in the abstract, specify the questionnaires and biometric tests used in the study.
Author’s Response: The methodology in abstract section is revised.
-“Study participant and setting” paragraph: are there any inclusion criteria based on age? the “setting” is not described (which hospital, etc.). on the other hand, questionnaires are not either “participant” or “setting”, so I suggest to put every test made on the study population in another paragraph. I also suggest to better explain the “study design”, i.e., for example: cross-sectional study assessing the level of kinesiophobia through questionnaires and the ability to move through biometrical assessments.
Author’s Response: No the age was not the inclusion criteria. Details of the hospital setting was described under the design heading. Now it’s provided under the settings heading.
-In Table 1, the first age of participants is 15 years old, although in the text it is specified that the study sample’s age ranged from 18 to 45.
Author’s Response: it was a copying error it’s 15-24, 25-34 & 35-45
-Line 135 is grammatically incorrect.
Author’s Response: The statement has been revised
-Line 153: “In comparison to the patients who had the ACL reconstruction surgery within the 153 year or after two years post-surgery “ - why post-surgery? please revise this phrase syntactically
Author’s Response: The statement was revised to make it more clear.
-Results: what about the RSI scale? what about the biometric tests specified in the “procedures” paragraph? if the results of these tests do not appear in the “results” paragraph, it is useless to describe them in the Methodology section.
Author’s Response: The ACL-RSI scale data was results were compared in the table 5. The items scores were also added. Additional details were removed from the methodology section.
-Discussion: “In this study, we are the first to discover a link between higher levels of Kinesiophobia and return to sport based on their functional performance disparities” (line 201). that is incorrect. the present study did not investigate anything related to functional performance - at least it was not explained in the Results section.
Author’s Response: The paragraph has been revised to make it specific to the scope of study.
- At the beginning of the manuscript you also cited the difference between a direct and an indirect injury to the ACL. I am curious whether you considered that piece of information in your study. does the amount of kinesiophobia increase according to the type of injury?
Author’s Response: It was in relation to the type of injury which happen during the sporting activities.

Reviewer 2 Report
I have only minor suggestions to authors.
1. table 1, please give the value (from to) for normal, overweight, obesity; age in year, please change to 15-24; 25-34.
2. Table 1 and 2, any statistical analysis? Chi-square
3. in the table 5, please use "r" or "R" to describe correlation. Please also check if Pearson correlation is adequate. Perhaps the better way was use Spearman rank ?
4. discussion, please add strengths of your work and future perspective, clinical implication from obtained results.
Author Response
Reviewers Comments and Suggestions for Authors
I have only minor suggestions to authors.
- table 1, please give the value (from to) for normal, overweight, obesity; age in year, please change to 15-24; 25-34.
Author’s Response: We have revised the mistake
- Table 1 and 2, any statistical analysis? Chi-square
Author’s Response: table 1 and 2 present descriptive analysis based on frequency and percent distribution for all patients’ demographic and injury related data. In table 3, bivariate relation between patients and injury data was tested using Mont Carlo Estimate and Exact probability test. A correlation analysis between Kinesiophobia level and return to sports scale after reversing the degrees to detect nature of relation.
- in the table 5, please use "r" or "R" to describe correlation. Please also check if Pearson correlation is adequate. Perhaps the better way was use Spearman rank ?
Author’s Response: As the ACL-RSI Scale is a measured on 0-100 scale Pearson correlation is adequate to showcase the relationship.
- discussion, please add strengths of your work and future perspective, clinical implication from obtained results.
Author’s Response: The relevant section in the discussion has been revised and few references have been added to accommodate the reviewers suggestions.

Round 2
Reviewer 1 Report
Thank you for your replies and corrections.